

# A review of organization-oriented phishing research

Kholoud Althobaiti and Nawal Alsufyani

Department of Computer Science, Taif University, Taif, Saudi Arabia

## ABSTRACT

The increased sophistication and frequency of phishing attacks that target organizations necessitate a comprehensive cyber security strategy to handle phishing attacks from several perspectives, such as the detection of phishing and testing of users' awareness. Through a systematic review of 163 research articles, we analyzed the organization-oriented phishing research to categorize current research and identify future opportunities. We find that a notable number of studies concentrate on phishing detection and awareness while other layers of protection are overlooked, such as the mitigation of phishing. In addition, we draw attention to shortcomings and challenges. We believe that this article will provide opportunities for future research on phishing in organizations.

## INTRODUCTION

Phishing is a devious and manipulative cybersecurity threat that targets organizations across all sectors worldwide (*APWG, 2023*), aiming at obtaining sensitive information, compromising systems, and securing financial gain. Organizations are faced with ongoing challenges in combating phishing because of the increase in the number of phishing attacks over the years along with continuous changes in attackers' strategies and tactics to get around security measures (*Cofense, 2023*). The implications of phishing attacks are damaging for organizations because they can lead to loss of revenue, reputations, and intellectual property with some of the attacks being used to initiate more complex cyber threats such as ransomware and data breach (*Verizon, 2022*). As a result, organizations have to defend against phishing attempts. They should adopt complete cybersecurity strategies, such as applying sophisticated technical defenses, educating staff members on security issues regularly, and creating guidelines and protocols for spotting and handling phishing attacks.

To enable organizations to fight against phishing, previous research studied phishing from several perspectives like developing interventions (*Franz et al., 2021*; *Goel & Jain, 2018*; *Singh & Meenu, 2020*), exploring human susceptibility to phishing (*Scott & Kyobe, 2021*), and analyzing phishing attacks (*T N, Bakari & Shukla, 2021*). Although these studies may provide a solid foundation on their own, combating phishing requires a multi-layer approach that focuses on applying several measures to make it harder for attackers to reach users and also implement all the reactive measures to reduce the damage and minimize its

Corresponding author
Kholoud Althobaiti,
kholod.k@tu.edu.sa

impact on organizations (*NSCS, 2018*). Therefore, exploring the diverse research landscape across various research disciplines can provide a holistic understanding of the fight against phishing in organizations. The rationale for this study arises from the increased number of such attacks. We believe that the systematization of existing literature would reveal the trend and gaps in the fight against phishing in organizations. Thus, our research aims to answer the following two questions:

**RQ1:** What are the current research directions targeting phishing in organizations?

**RQ2:** What are the open questions in the current literature for future research?

In this work, we provide a comprehensive systematization of organizational-oriented phishing research in terms of the process of handling phishing attacks in organizations from the first steps taken to protect organizations against phishing to the mitigation steps and learning from a successful attack (*SecAware, 2013*). We analyzed a total of 163 research papers published between 2012 and 2024 based on predetermined criteria, such as the use of organization-based keywords to ensure the study captures the spectrum of the organizational fight against phishing. The main contribution is the analyses of the existing literature which resulted in categorizing the studies based on the type of study and layer of protection. We found that the literature is heavily based on detecting phishing attacks and raising users' awareness with a limited number of studies on post-attack studies such as phishing incident response. This research is targeting cybersecurity professionals, researchers, and organizational decision-makers who are directly involved in the fight against phishing attacks.

The remainder of the article is organized as follows. First, we present the type of literature reviews done in the area along with a short background on the phishing life cycle. Second, we discuss the methodology adopted for this study followed by the results of the study. We then discuss the results and conclude the article.

## BACKGROUND

To situate our research in the literature, this section discusses the previous review papers and taxonomies and explains a high-level description of the phishing life cycle that involves attack and defense in organizational settings.

### Phishing taxonomies

Various taxonomies and classification schemes have been proposed to structure research on phishing attacks. One common phishing taxonomy considers the medium of the attack, target environment, and tactics by *AlEroud & Zhou (2017)* where *Goel & Jain (2018)* focused their review specifically on mobile phishing attacks as one common medium of phishing attacks. Social engineering attacks have been categorized based on their type: phishing, pharming, and spoofing (*Mathew, Al Hajj & Al Ruqeishi, 2010*). The mechanism-based review involves distinguishing between social engineering (*e.g.*, spam and phishing) and technical subterfuge (*e.g.*, impersonation) (*Gupta, Arachchilage & Psannis, 2018*), which is also can be categorized into conventional and automated techniques depending on the attacker's technicality (*Qabajeh, Thabtah & Chiclana, 2018*). *Gupta, Arachchilage & Psannis (2018)* proposed a taxonomy of various methods used to protect users from

technical and non-technical phishing attacks, which is categorized into user education, and automated detection of emails and websites. Automatic detection of phishing emails has been a popular area of study (*Muneer et al., 2021*), with some reviews specifically examining techniques such as machine learning-based detection (*Singh & Meenu, 2020*; *Gangavarapu, Jaidhar & Chanduka, 2020*), deep learning (*Dixit & Silakari, 2021*), and natural language processing (*Salloum et al., 2022*). Another study focuses on the countermeasures for a specific type of phishing, business email compromise (BEC), by categorizing its techniques and countermeasures (*T N, Bakari & Shukla, 2021*).

Additionally, user-centered phishing research has been approached from several perspectives. One study explored the literature to understand the characteristics and traits of phishing victims (*Darwish, Zarka & Aloul, 2012*), while another reviewed the human factors in phishing attacks (*Desolda et al., 2021*). User-centered interventions were examined by *Franz et al. (2021)*, analyzing existing approaches, attack vectors, and types of user interaction. Moreover, *Jampen et al. (2020)* and *Aldawood & Skinner (2018)* explored the effectiveness of security awareness programs in raising users' awareness.

The above reviews encompass general research on phishing that is targeted at both individuals and organizations. In our review, we focus on categorizing the literature on organizational-based research.

## Phases of phishing attacks

A phishing attack is a social engineering attack that involves multiple phases of activities: the pre-attack phase, the attack phase, and the post-attack phase (*Alizadeh et al., 2023*). These phases may not occur only once but instead, they appear in a cyclical pattern.

The pre-attack phase involves activities aimed at exploring their goal, identifying a target, and learning more about them. Following the information gathering, the attacker can plan the strategy, technique, and the appropriate channel to achieve the goal (*Gupta et al., 2017*; *Thurman, 2020*; *Oest et al., 2018*; *AlEroud & Zhou, 2017*; *Alabdan, 2020*).

The attack phase includes the execution of the attack itself, which entails establishing communication and interaction with the target user and building a relationship. This process requires fabrication, such as impersonation and the use of false identifiers, to deceive the target (*Purkait, 2012*; *AlEroud & Zhou, 2017*) and persuade them to comply with the attacker to fall victim to the attack (*Parsons et al., 2015*; *Rader & Rahman, 2015*; *Benenson, Gassmann & Landwirth, 2017*).

The post-attack phase covers the exploitation of trust, the use of obtained information (*e.g.*, passwords or card details), or the exploitation of security vulnerabilities (*e.g.*, malware) at the appropriate time (*AlEroud & Zhou, 2017*; *Mouton et al., 2014*). This phase also includes covering tracks, which involves deleting event logs, fake accounts, or fake websites to remove evidence of the attack (*Shaikh, Shabut & Hossain, 2016*). It may also include blocking the user's access to their account by changing their passwords (*Qabajeh, Thabtah & Chiclana, 2018*; *Steer, 2017*; *Bursztein et al., 2014*; *Onaolapo, Mariconti & Stringhini, 2016*). The post-attack phase also involves ensuring the victim feels safe without noticing any suspicious activities and ensuring that the attackers achieved their goal (*Mouton, Leenen & Venter, 2016*).

## Phases of phishing protection

Learning how attackers establish and execute phishing attacks helps organizations adopt and adapt frameworks and best practices (*Frauenstein & von Solms, 2009*; *Frauenstein & von Solms, 2013*; *Hammour et al., 2019*; *Moul, 2019*) to implement thorough countermeasures for protecting their assets and users. These measures encompass both proactive defenses, which are implemented before an attack, and reactive measures, which are taken during or after the attack.

Organizations primarily focus their proactive efforts on the early detection of phishing, often achieved through employee training and preventive technological measures (*Kokulu et al., 2019*). Some preventive technical measures include blocking emails before they reach the mail server (*Purkait, 2012*; *Park et al., 2014*) or blocking websites when a user's browser requests malicious content (*Frauenstein & von Solms, 2009*; *Tsalis et al., 2014*; *Jain & Gupta, 2016*). While these measures help detect phishing incidents, they cannot guarantee complete accuracy, necessitating concentrating efforts on training users to identify threats that may bypass these filters.

Cybercriminals continually seek new methods to evade proactive defenses, compelling organizations to establish reactive procedures for addressing new attacks. These reactive measures involve monitoring automated alarms, tracking user-reported phishing attempts, and scanning for suspicious activities (*Arachchilage & Cole, 2016*; *Abawajy, 2014*; *Arachchilage, Love & Maple, 2015*). Timely responses significantly aid organizations in responding to and minimizing the impact of attacks, thereby reducing the likelihood of potential victims engaging in phishing communications and mitigating harm to the organization.

While this provides an overview of the phishing life cycle, there is limited knowledge about the research conducted in the area including attacker strategies, users' awareness, interventions, challenges encountered during any of the phases, and incident handling within the organizational environment, specifically within information security centers.

## METHODOLOGY

To understand the current landscape of phishing research focused on organizations, a systematic literature review was performed by following the guidelines of *Okoli (2015)*. Literature reviews are essential for advancing domain knowledge as they synthesize previous research and identify research gaps.

### Selecting literature

Phishing is a highly prevalent topic that crosses multiple fields, resulting in papers published across various fields, including human–computer interaction (HCI), computer security, cryptography, and information systems. Consequently, three digital libraries were selected for the search: Association for Computing Machinery (ACM) Digital Library, Institute of Electrical and Electronics Engineers (IEEE) Xplore, and ScienceDirect.com, which cover the majority of relevant fields. The search keywords were consistent across databases and applied to titles, abstracts, or author-specified keywords (refer to Appendix for the queries used for each library).

In selecting keywords for the literature search, we focused on terms commonly found in recent studies on organizations. We initially experimented with a range of both broad and specific keywords to capture relevant research. This process was essential, given the extensive volume of phishing-related literature, which would otherwise result in an enormous number of articles to review. To be considered, articles had to include the term 'phish' or 'phishing' and one of the terms 'organisation', 'organization', 'institution', 'corporation', 'enterprise', 'workplace', or 'incident'.

## Inclusion criteria

Using the search method outlined above, a total of 620 publications were initially identified, with some appearing in at least two libraries. The search was conducted on April 13, 2022. We then restricted the selection of articles from 2012 onwards to ensure the inclusion of recent research relevant to current challenges and advancements.

Extended abstracts, posters, and literature review papers were excluded to focus on peer-reviewed articles to focus on original research and empirical studies. We also limited the review to papers written in English.

After the initial exclusion and removal of duplication and unavailable 'pdf' files, 544 articles remained. The authors then commenced the full-text screening process by individually reading and analyzing the articles based on the following inclusion criteria: We included research specifically targeting or investigating organizational settings. More specifically, studies had to be tested in organizational environments or use data collected from specific organizations to ensure practical applicability. While many articles focused broadly on cybersecurity, we included only those that discussed phishing within the context of broader cybercrimes if they had dedicated sections addressing phishing specifically. Studies focusing on methods to prevent or remediate phishing, such as authentication mechanisms and anomaly detection systems, were also included to align with the study's objectives.

The full-text analysis further reduced the literature count by 105 articles.

The authors held several meetings during the screening process to ensure thoroughness and consistency in applying the inclusion criteria. These meetings helped to minimize biases and priming effects where discrepancies in article inclusion decisions were resolved through discussion and consensus to ensure reliability and validity in the selection process.

After writing the article, we researched the literature on June 10, 2024, to include any missing papers. The same screening process was applied to the new literature, resulting in 58 additional studies, increasing the total number of studies reviewed to 163 studies. Figure 1 summarizes the entire process of selection of publications.

## A TAXONOMY OF ORGANIZATION-BASED PHISHING RESEARCH

While reviewing the literature, we observed a variation in themes concerning the underlying stage of the attack and defense life cycle. After several discussions among the researchers, we chronologically categorized the literature starting with the proposal of policies and frameworks and then the factors that affect the susceptibility to phishing, testing of user

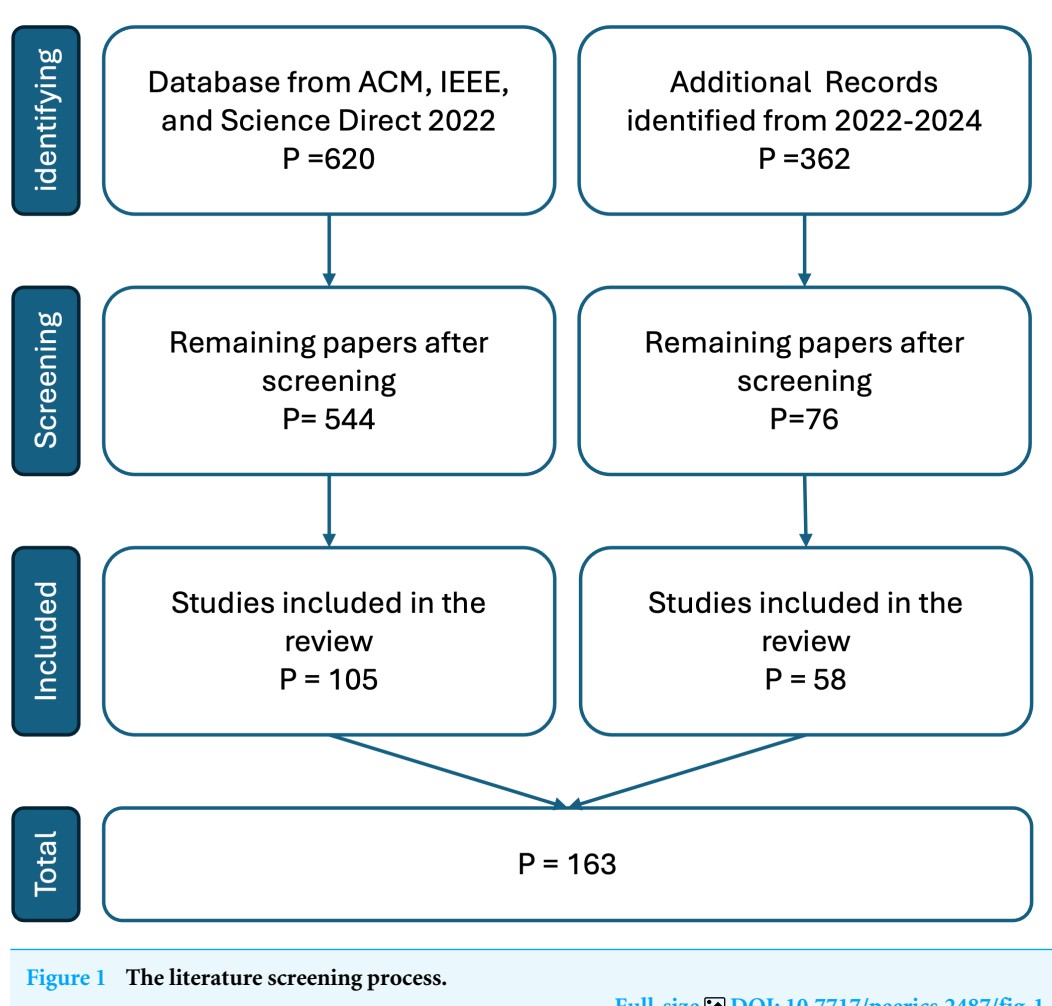

**Figure 1** The literature screening process.

awareness, and interventions used to raise their awareness, the detection of phishing communications, and research on incident response to phishing attacks, find the list of categories in Fig. 2 and a summary of the findings in Table 1.

## Policies and frameworks

Organizations are required by law to comply with the government-imposed cybersecurity regulations, which are found to raise organizations' awareness of threats and positively affect companies' decisions to invest in IT and security, as seen in Michigan and Oregon organizations (*Wang et al., 2024*). Therefore, well-established cyber security frameworks such as COBIT, NIST, and ISO27001 complement the regulations by offering a comprehensive approach.

Prior research also proposed policies and frameworks to defend the organization's ecosystem specifically from phishing attacks. As an example suggesting organizational compliance with training employees, educating them, and banning the sharing of passwords and sensitive information (*Itani et al., 2024*). One of the studies focuses its framework on the relation between key elements attackers aim to exploit: human factors, organizational

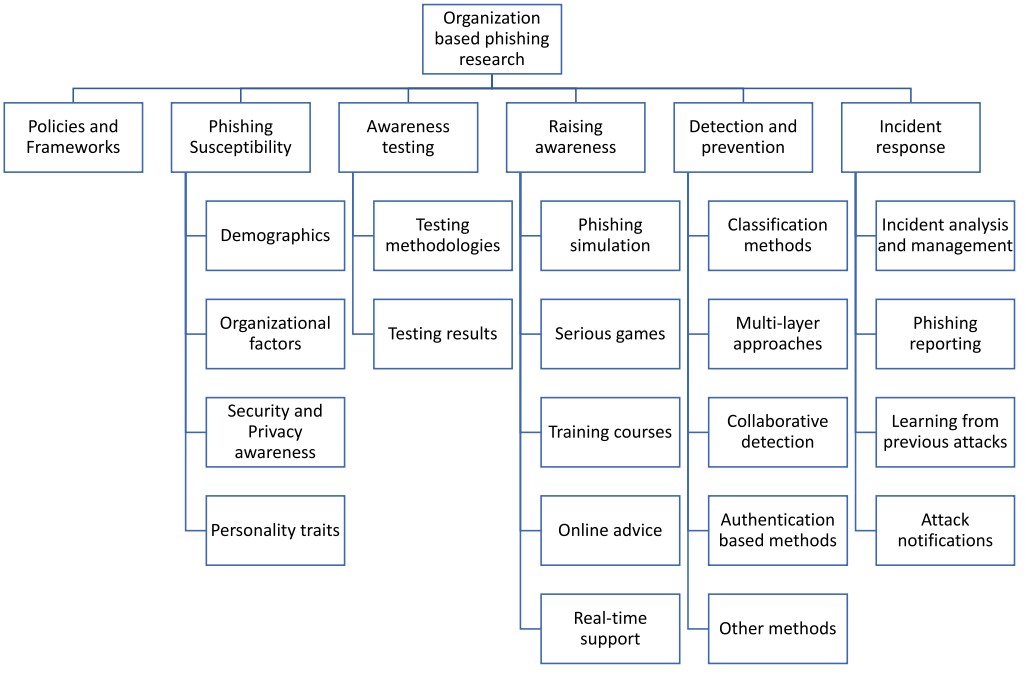

**Figure 2** The resulted taxonomy of organization-based research.

aspects, and technological controls (*Frauenstein & von Solms, 2014*). This relationship involves the use of best practices such as COBIT to identify suitable technological techniques, the assurance of adequate policies and procedures to dictate the employees' behavior, and teaching employees how to use the organizations' technological controls. Whereas another study focused only on the human factor element by proposing an AI-based user and entity behavior analytics framework that helps analysts assess each user's exposure to threats based on real-world data (*Calvo et al., 2023*). This approach identifies potential vulnerabilities before attacks happen and allows organizations to prioritize their security efforts and implement preventive measures.

The size of the organization plays an important role in the framework used because small and mid-sized organizations, although targeted by phishing attacks, they have limited budgets with restricted resources, requiring them to size up their budget to the needed security investment. *Rodríguez-Corzo, Rojas & Mejía-Moncayo (2018)* proposed a phishing model that focuses on three actors: business, technology, and people. For the business factor, the business characteristics, threats, and the attacker purpose should be identified because they will be utilized in the technology factor by identifying its current status and resources needed to implement the model. The third actor is people who are the training staff and their exposure to risk based on their position.

Other studies focus their framework or policy on specific types of phishing attacks or vulnerabilities. *Shakela & Jazri (2019)* proposed a Spear Phishing Exposure Level (SPEL) framework to assess and reduce spear phishing-related threats, which is designed from two perspectives: the threat source and vulnerability to threats. This approach allows

**Table 1 Summary of organization-based phishing research.**

| Category | Subcategory | Findings |
| --- | --- | --- |
| Policies and frameworks | | - Cybersecurity law & frameworks influence organizational awareness of security. |
| | | - Policies focus on training, data protection, and vulnerability checks. |
| | | - Organization size plays a role in cybersecurity decisions. |
| Phishing susceptibility | Demographics | - While age and major is a susceptibility factor, gender has no impact. |
| | Organizational factors | - Overconfidence in security and distractions increase risk. |
| | | - Tenured employees, managers, and tech staff have higher awareness. |
| | | - Level of awareness differs between sectors |
| | Awareness | - Security awareness and knowledge reduce risk. |
| | Personality traits | - High conscientiousness and low agreeableness and trust reduce vulnerability. |
| | | - High Influence decreases risk compared to high Dominance. |
| Awareness testing | Methodology | - Phishing simulation is most common compared to surveys. |
| | | - Choice of the methodology is cultural-dependent. |
| | Results | - High success rates of phishing attacks requiring better training. |
| | | - Targeted simulations reduce susceptibility and increase reporting. |
| Raising awareness | Phishing simulation | - Limited effectiveness for users already vulnerable. |
| | | - Challenges include data privacy and management in large organizations. |
| | Serious games | - Employ personality traits, role-playing scenarios, and quizzes. |
| | | - Awards and sanctions increase engagement. |
| | Training courses | - Commonly integrated into curricula and IT programs. |
| | | - Combining theoretical training with other methods increases effectiveness. |
| | Online advice | - Social networks can be used as source for online advice. |
| | | - Public anti-phishing web pages lack clear, concrete advice. |
| | Real time support | - Security nudges and extensions improve phishing detection. |
| | | - user-centric tools are needed for real-time support. |
| Detection and prevention | Classification methods | - Common methods are machine learning, deep learning, clustering, and NLP. |
| | Multi-layer methods | - Multi-layer integration increases the classification accuracy. |
| | | - Additional layers include IP mapping, logo recognition, and dynamic firewall rules |
| | Collaborative detection | - Information sharing between organizations can improve defense measures. |
| | | - Privacy-preserving protocols for secure data queries. |
| | Authentication | - Email authentication protocols help but have a low adoption rate |
| | | - Dynamic passwords, brand indicator authentication, and nonce message protocols enhance phishing prevention. |
| | Other methods | - Various methods such as detecting bogus invoices, QR code manipulation, malicious executable files, ransomware, and lateral phishing. |
| | | - Virtual honeypots to block access to phishing sites. |
| | | - Detection of encrypted phishing sites, domain-based brand protection, and multi-stage site takedown enhance security. |

**Table 1** (*continued*)

| Category | Subcategory | Findings |
|---|---|---|
| | Attack insights | -Frequent analysis of attacks enhances protection against increasingly sophisticated attacks. |
| Incident response | Notifications | - Notification about ongoing attacks has limited effect. |
| | Incident management | - Incidents analysis tools increase situational awareness. |
| | | -Attack modeling techniques enhance risk assessment and response strategies. |
| | Reporting attack | Phishing reporting enhances response time and resilience while reducing security costs. |

organizations to determine the possibilities of attack in the absence of security mechanisms. *Ismail et al. (2017)* proposed a policy to mitigate the impact of watering attacks and spear phishing; both allow the attacker to take control of an insider account. After surveying business stakeholders, the authors found that role-based access control is not enough if the attack originates from a compromised insider. Therefore, they first proposed adding another access control layer, such as Bell La-Padula, which uses access labels on objects and clearances for subjects to reduce the impact of unauthorized access. Then they proposed policies include setting complex passwords, reporting lost devices, disabling the ability to install apps from a third party, enforcing the installation of anti-viruses and firewalls on personal devices, disallowing the use of jailbroken or rooted personal devices to connect to organization facilities and only allowing activated accounts to be connected *via* wireless. To evade phishing attacks in traditional tender allocation systems, *Dubey et al. (2023)* proposed using blockchain along with common security practices through smart contracts and immutable records, which contribute to ensuring fair competition by automating the tender process from creation to winner selection. For bring-your-own-device environments, *Bann, Singh & Samsudin (2015)* propose a multi-level security and access control policy based on the analysis of four well-established security policies namely, mandatory access control (MAC), Clark Wilson, Low Water Mark access mandatory control (LOMAC), and attribute-based access control (ABAC) based on various quality metrics, such as the size of the organization, the cost of administration, and complexity. To evaluate the applied cyber security frameworks in defending against phishing, *Kulkarni et al. (2024)* suggested the use of tools such as *HiddenEye* and *sendemail* in a simulated environment can asses with phishing countermeasures.

## Phishing susceptibility

Crafting phishing attacks often relies on a common method that is referred to as social engineering, which uses psychology and behavior to make emails seem more trustworthy and discourages people from carefully checking the information they receive. This approach increases the probability of successful attacks. While the previous theme examined attacker strategies used in real attacks, in this theme, we discuss the research that addresses the technical and human factors related to phishing susceptibility.

*Demographics.* Studies show that demographics are a significant factor in determining individuals' susceptibility to phishing attacks. A study of university students in Emirates showed that junior students are less likely to fall victim to phishing than senior

students (*Mohebzada et al., 2012*). Likewise, an investigation of multi-national financial organizations revealed that older participants are more vulnerable to phishing than younger participants (*Taib et al., 2019*). This observation was ascertained in several companies from various sectors (*Lain, Kostiainen & Čapkun, 2022*); however, this correlation was not observed with users of privacy-preserving technology companies (*Clark, 2012*). Younger users are more susceptible at the early stages (receiving and opening emails), whereas older users are more likely to fall victim at advanced stages (*e.g.*, visiting phishing sites) (*Zhou, Zhang & Liu, 2023*). Considering the variation in studies' results, we can attribute this to differences in participants' age ranges, with some studies focusing on ages 18 to 25, while others include participants up to 40.

On the other hand, empirical studies have found that gender does not have a statistically significant impact on one's likelihood of falling victim to an attack (*Mohebzada et al., 2012*; *Taib et al., 2019*; *Flores, Farid & Samara, 2019*; *Clark, 2012*; *Zhou, Zhang & Liu, 2023*; *Lain, Kostiainen & Čapkun, 2022*; *Ribeiro, Guedes & Cardoso, 2024*) except for a study in a Philippine university showing higher awareness of spam (*Hermogenes & Capariño, 2019*). Thus, while age remains a crucial factor in identifying those at greater risk, gender does not appear to influence susceptibility to phishing.

Regarding the university major, studies showed that IT students have a higher awareness of phishing emails and cybercrimes compared to their peers in education and science fields (*Manasrah, Akour & Alsukhni, 2015*); likewise, they have the best resistance against phishing attacks (*Andrić, Oreški & Kišasondi, 2016*). Similarly, 54.32% of IT students were aware of spam and phishing compared with students from teaching education programs (*Hermogenes & Capariño, 2019*), signifying the importance of technology competencies in phishing susceptibility (*Ribeiro, Guedes & Cardoso, 2024*). This finding is contrasted by *Clark (2012)*, who reported an absence of correlation between an employee's field of study in the US, whether computer-related or not, and their susceptibility to phishing attacks in experimental settings. These contrasting results highlight a complex landscape of cyber literacy and vulnerability among university students, suggesting that awareness does not necessarily equate to immunity from cyber threats.

*Organizational-based factors.* Several key elements are found in the analyzed literature. The strong belief in organizations' security measures may lead to a lower sense of risk (*Kearney & Kruger, 2014*) as observed in interviews with banking IT staff who assumed that their security protocols would prevent attacks or quickly rectify its implications (*Conway et al., 2017*).

Additionally, the length of time employees had worked at the company played a role. Newer employees, especially those in their first year, were found to be more vulnerable to phishing attempts than their more tenured colleagues (*Kearney & Kruger, 2014*). This finding was supported by a large-scale simulation in financial organizations (*Taib et al., 2019*) and a field experiment at a university and a large international consultancy company (*Burda et al., 2020*).

Furthermore, job responsibilities influenced vulnerability to phishing; thus, employees with managerial duties tended to be more cautious, potentially due to their investment in

the company's image or the consequences of non-compliance (*Taib et al., 2019*; *Eftimie et al., 2021*). Additionally, employees with higher effective organizational commitment show higher awareness in various Australian organizations (*Reeves, Parsons & Calic, 2020*). For example, employees from technology-based departments have a higher level of awareness than those from social-based departments in a Thai organization (*Daengsi et al., 2021*). Also, job roles that utilize a centralized inbox lead to increased exposure to potential phishing emails due to the nature of such inboxes (*Williams, Hinds & Joinson, 2018*). Apart from the job roles, generally, the frequent use of general computers and usual internet routine reduce employees' susceptibility to phishing (*Ribeiro, Guedes & Cardoso, 2024*; *Lain, Kostiainen & Čapkun, 2022*).

The susceptibility to phishing was also examined between sectors. Government employees have more knowledge of phishing protection than employees from private sectors in Saudi Arabia (*Innab et al., 2018*). Adding to that, employees associated with privacy-preserving technology companies in the US are still likely to disclose personal information in phishing scenarios (*Clark, 2012*).

Interruptions during tasks increase the likelihood of employees falling for phishing scams, as they may be less focused and thus more susceptible to fraudulent communications (*Williams, Morgan & Joinson, 2017*).

The importance of organizational norms in influencing employees' compliance with information security policies (ISPs) has been highlighted by *Petrič & Roer (2022)* and *Williams, Hinds & Joinson (2018)*. The study provides comprehensive knowledge of how many normative variables, in particular phishing vulnerability, affect employees' conduct. Notably, the investigation shows that the effect of descriptive norms on the tendency of staff members to click on questionable links differs from the effect of personal norms in the same tendency. It has been shown that workers who adopt security-promoting norms are less likely to interact with phishing emails and to exercise caution when clicking on embedded links. The research suggests that a moral commitment to organizational security norms triggers more analytical processing of emails. However, this commitment may not always safeguard against sophisticated phishing tactics post-click.

These results imply that evaluating and mitigating phishing susceptibility in an organizational setting requires a comprehensive approach that takes into account variables including job role, tenure, perceived security strength, and organizational norms.

*Security and privacy perception and knowledge.* The perception and knowledge of security and privacy can play a crucial role in an individual's susceptibility to cyber threats. A study of Middle Eastern countries revealed that the perceived high-security risk does not always translate to protective actions as the participants may fall prey to phishing (*Aleroud et al., 2020*). Similarly, in Australia, individuals with a lower fear of cyber threats exhibited better information security awareness (*Reeves, Parsons & Calic, 2020*; *Ribeiro, Guedes & Cardoso, 2024*).

Furthermore, a lack of security knowledge, such as misunderstanding security indicators on websites, leaves users more open to deception by phishers who often exploit such gaps in knowledge (*Aleroud et al., 2020*; *Williams, Hinds & Joinson, 2018*). Knowledge of the

difference between HTTP and HTTPS and URL syntax and shorteners can help university students avoid suspicious emails (*Andrić, Oreški & Kišasondi, 2016*). A lack of technical understanding about what spear phishing entails, how personal information is used in attacks, and the consequences of engaging with phishing emails can lead to increased susceptibility (*Williams, Hinds & Joinson, 2018*), especially for financial consequences (*Aleroud et al., 2020*). Knowledge of red flags in phishing emails, such as spelling errors and sender address is critical (*Williams, Hinds & Joinson, 2018*; *De Bona & Paci, 2020*; *Buckley et al., 2023*). While this knowledge can be effective and used by several users, they might be misleading in the occurrence of lateral phishing, requiring knowledge of the sender's writing style and the expectation of communication topic (*Chitare, Coventry & Nicholson, 2023*).

Higher privacy behavior reduces individuals' susceptibility to phishing (*Zhou, Zhang & Liu, 2023*), particularly among women in the Middle East, influencing their willingness to share personal details; however, this caution does not necessarily extend to protecting them from phishing (*Mohebzada et al., 2012*; *Aleroud et al., 2020*).

Although raising knowledge of security and privacy is critical, it does not always protect against phishing, signifying the need for improving security procedures and education.

*Personality traits.* The Big Five personality traits have been linked to various behaviors concerning vulnerability to phishing. Highly conscientious individuals typically engage in responsible security practices, whereas those with lower levels of conscientiousness are prone to riskier behaviors (*Eftimie et al., 2021*). Similar to the findings above, those in leadership positions, often characterized by low levels of agreeableness and high levels of conscientiousness, are less likely to fall for phishing attacks (*Eftimie et al., 2021*; *Yaser Al-Bustani et al., 2023*). Regarding trust, it significantly affects vulnerability to spear phishing; especially in Middle Eastern countries. Phishers usually exploit trust, particularly through credible-looking and contextually convincing materials, which underline the intricate link among personality, trust, and susceptibility to phishing (*Aleroud et al., 2020*). Experimenting with financial organizations shows that employees with higher trust in their intuition are less likely to engage with phishing emails (*Buckley et al., 2023*). In addition, individuals' self-efficacy in detecting phishing attempts demonstrated a significant influence on their susceptibility to phishing attacks (*Ribeiro, Guedes & Cardoso, 2024*). People with high Influence are less susceptible to phishing due to their social awareness and caution, those with high Stability are moderately resistant but still have some vulnerabilities, while individuals with high Dominance are more susceptible because they prioritize results over careful scrutiny (*Yaser Al-Bustani et al., 2023*).

## Phishing awareness testing

Awareness testing in organizations is a critical component of cybersecurity strategy, particularly in addressing the susceptibility of individuals to phishing attacks. This testing is conducted using a variety of methodologies, each revealing distinct aspects of human vulnerability and behavioral tendencies in the context of cyber threats.

*Testing methodology.* To assess and enhance awareness, previous research employed various methodologies. The most common methodology is the *phishing simulation experiments* (*Bakhshi, 2017*; *Blancaflor et al., 2021*; *Bakar, Mohd & Sulaiman, 2017*). An example is sending an email asking the employees to click on a link to a survey (*Bakhshi, 2017*; *Blancaflor et al., 2021*). The success of these simulated attacks can reveal the extent of vulnerability among the participants. To boast the benefits of such methodology, *Rutherford, Lin & Blaine (2022)* utilized the machine learning algorithms to understand the simulation results based on the potential victim's demographics and administrative data.However, setting up simulated phishing experiments to measure actual behaviors is not only expensive but also raises ethical concerns, such as user consent.

Surveys developed to gather information can precede simulated phishing attacks. For example, researchers in Manila collected personal details through a survey and then used this information to launch targeted phishing attacks (*Blancaflor et al., 2021*). In some cases, surveys can effectively replace simulation methodologies. The results of an experiment in an Indonesian government sector revealed a significant relationship between the simulation results and the questionnaire results (*Ikhsan & Ramli, 2019*). However, *Flores et al. (2015)* suggested that the methodology used is dependent on the culture as surveys can be used as a proxy to measure employees' intention to avoid social engineering in Sweden while scenarios are the best proxy in American culture, indicating that the use of assessment methods can differ between national cultures. Another method to test users' awareness is the use and development of standard scales. A phishing experiment with Australian students revealed that students who achieved high scores in the experiment also achieved a high score on the Human Aspects of Information Security Questionnaire (*Parsons et al., 2017*).

*Testing results.* Several studies have highlighted how surprisingly easy it can be to execute successful phishing attacks within organizations. For instance, in a branch office of a cooperative organization, a significant number of employees were comfortable sharing sensitive information, such as details about office supplies and equipment (*Bakhshi, 2017*). This finding underscores a lack of awareness regarding the sharing of potentially sensitive information. In a mid-sized university, 44.3% of users clicked on at least one phishing email, with 18.6% entering valid credentials (*Cuchta et al., 2019*); similarly, 42% of students in another experiment visited and completed forms and downloaded the email attached image (*Rastenis et al., 2019*). In total, 38% in a Malaysian university also entered their work ID and password, and 95% agreed to receive the financial aid (bait) (*Bakar, Mohd & Sulaiman, 2017*). This pattern of failure is not that different in surveys where 25% of students failed to correctly identify phishing emails in the survey (*Andrić, Oreški & Kišasondi, 2016*). A long-term study in various sectors revealed that about 32% will fall for phishing at least once if exposed to phishing emails (*Lain, Kostiainen & Čapkun, 2022*). These high engagement rates with phishing attempts indicate a substantial gap in awareness and the ease with which attackers can exploit this vulnerability. Studies in higher education institutions have shown that a large portion of students and employees are influenced by

phishing emails, with a notable percentage of them providing personal data. This finding suggests a need for enhanced security education and training.

## Raising users awareness

The field of raising phishing awareness in organizations has seen extensive research, addressing various methodologies and their effectiveness such as instructor-led, video-based, text-based, and game-based training along with real-time support. Although the delivery approach is important, *Alkhazi et al. (2022)* observed that the enjoyable training sessions encourage Kuwaiti government employees to engage in self-learning and future training. To design the awareness materials, most of organizations rely on experts to tailor the content but some use non-expert crowd-sourcing participants to identify the common phishing cues that can be used in training based on the recent phishing attacks. This method helps provide training from the perspective of the system end-users capable of providing fresh, diverse, and comprehensive phishing cues over time (*Rosser et al., 2022*).

*Phishing simulation for training.* Simulated phishing exercises are also used to train employees to recognize and respond to phishing attempts. This method involves creating and sending simulated phishing emails to employees, which mimic the tactics and appearance of real phishing emails, but without the malicious intent. The goal is to expose employees to the types of phishing that they might encounter in a safe and controlled environment. Such studies were explored in several sectors and countries such as transportation in Bangkok (*Sirawongphatsara et al., 2023*), various small Japanese organizations (*Higashino et al., (2019)*, healthcare sector (*Williams, Zafar & Gupta, 2024*), and Israeli financial institution (*Hillman, Harel & Toch, 2023*). A notable finding across several studies, including research in Italy (*De Bona & Paci, 2020*) and the USA (*Pirocca, Allodi & Zannone, 2020*), shows that targeted simulated phishing training reduces susceptibility compared with generic phishing training (*McElwee, Murphy & Shelton, 2018*) with a noticeable increase in phishing reporting rate (*Hillman, Harel & Toch, 2023*). However, analysis of mid-sized and large companies demonstrates this method's limited effectiveness for those already susceptible to phishing (*Siadati et al., 2017*; *Lain, Kostiainen & Čapkun, 2022*). Other research explored combining the phishing simulation with rewards and sanctions in organizations to mitigate risky behavior (*Blythe, Gray & Collins, 2020*; *McElwee, Murphy & Shelton, 2018*).

Although this type of training is effective in raising awareness, it is challenging to run; for example, using the open-source framework raises concerns about exposing staff information suggesting storing users' data locally rather than using public servers (*Higashino et al., 2019*). If run locally without tools, it is overwhelming to manage in large organizations (*Althobaiti, Jenkins & Vaniea, 2021*).

*Serious games.* Serious games are typically employed in various fields. Serious games include interactive games for education and training. *Pantic & Husain (2018)* applied the Five-Factor Model of personality traits to correlate types of phishing emails with individual vulnerabilities, suggesting a more personalized approach for training. *Underhay, Pretorius & Ojo (2016)* proposed a game-based e-learning model for university technology students

in South Africa. The game involves role-playing as a system administrator, requiring players to secure networks and systems. *Gupta et al. (2020)* developed a serious game for cybersecurity professionals to identify sophisticated phishing emails using a mix of quizzes and feedback mechanisms. Similarly, *Birajdar & T N (2022)* developed a serious game for IT professionals that concentrates on aspects such as interactivity, depth of knowledge, awards, and sanctions.

*Phishing training courses.* Phishing training is used for various purposes, such as educational curricula and IT training programs. *Turner & Turner (2019)* integrated phishing awareness modules into social studies classes, demonstrating positive outcomes in understanding and preventing phishing attacks. Interactive tools and apps are used to provide training in order to increase awareness among students and professionals in American high schools (*Podila et al., 2020*) and various sectors in Qatar (*Al-hamar & Kolivand, 2020*). This type of training is also used for helping cybersecurity students efficiently create spear phishing attacks, such as developing the Social Engineering Vulnerability Evaluation (SiEVE) process, a method for identifying targets, profiling them, and crafting highly personalized social engineering attacks (*Meyers et al., 2018*).

Combining theoretical training with practical training in a medical organization in the Slovak Republic improved phishing awareness to 13% (*Madleňák & Kampová, 2022*). Additionally, combining two or more training methods can effectively enhance security awareness, such as providing text-based training along with gamification (*Alkhazi et al., 2022*). For example, due to the limited resources in small-sized universities, *Matovu et al. (2022)* incorporated the in-class lectures for training combined with after-training Kahoot!-based games.

However, even after employing security training, a significant proportion of employees are still vulnerable to phishing (*Kearney & Kruger, 2014*; *Madleňák & Kampová, 2022*), indicating the need for long term awareness plan.

*Online phishing advice.* The use of Twitter-based awareness strategies for bank customers in the Emirates revealed that while there is increased use of social media for fraud awareness, the impact and clarity of the advice vary (*Skula, Bohacik & Zabovsky, 2020*). *Mossano et al. (2020)* analyzed the publicly available anti-phishing web pages and found a lack of inconsistencies and concrete advice (*Mossano et al., 2020*).

*Real time support.* Previous research examined the impact of providing support to users when they encounter potential phishing attempts. The use of security nudges (*e.g.*, highlighting the sender) improves individuals' detection of phishing emails (*Nicholson, Coventry & Briggs, 2017*). Similarly, using the EyeBit extension that deactivate all the forms inputs if the user did not look at the address bar underscores the importance of real-time, user-centric tools in combating phishing (*Miyamoto et al., 2014*).

The real-time support that is coming from peers and family was studied by *Coronges et al. (2012)*. They studied the impact of social networks on mitigating the spread of phishing attacks, investigating whether warnings from friends or superiors are more effective in preventing successful phishing incidents. However, highly central individuals did not warn

others about phishing attacks, meaning these networks are ineffective. Adding a warning on the top of a suspicious email as real-time support intervention significantly reduces phishing clicks and dangerous actions, though the length of the warning has no significant difference in the click rate (*Lain, Kostiainen & Čapkun, 2022*).

## Phishing detection and prevention

Previous research has explored a diverse range of approaches and technologies to combat phishing attacks using various attack vectors and methods, aiming at improving the organizations from phishing attacks across different vectors including websites (*Oest et al., 2019*; *Cuzzocrea, Martinelli & Mercaldo, 2018*; *Chen et al., 2021*; *Aslam & Nassif, 2023*), emails (*Stembert et al., 2015*; *Sanchez & Duan, 2012*; *Vos, Erkin & Doerr, 2021*; *Lam & Kettani, 2019*; *Lee et al., 2021b*; *Zeng, 2017*), voice phishing (*Yu et al., 2024*), and job advertisements posted on a popular Australian platform (*Mahbub, Pardede & Kayes, 2022*), with some of these studies focus on accuracy against specific characteristics such as languages (*Dunder, Seljan & Odak, 2023*; *Yu et al., 2024*). Notably, the approaches discussed concentrate on protecting organizations, utilizing organizational data, and improving the usability of phishing detection systems, such as using an interactive website to ease scanning and enhance the classifier accuracy (*Shombot et al., 2024*).

*Classification based measures.*  Classification-based methods are widely employed for detecting phishing attacks. These methods leverage various classification techniques such as machine learning (*Mahbub, Pardede & Kayes, 2022*), deep learning (*He et al., 2024*; *Devalla et al., 2022*), natural language processing (*Dunder, Seljan & Odak, 2023*; *Tudosi et al., 2023*), computer vision (*Pires & Borges, 2023*). These classification methods utilize a wide range of features such as URL-based features (*Swarnalatha et al., 2021*; *Devalla et al., 2022*; *Bouijij, Berqia & Saliah-Hassan, 2022*; *Aslam & Nassif, 2023*), image-based features extracted from websites screenshots (*Tanimu & Shiaeles, 2022*), websites cost features (*Ito, Takata & Kamizono, 2022*), and social features extracted from LinkedIn profiles (*Dewan, Kashyap & Kumaraguru, 2014*). For detecting voice phishing calls, *Yu et al. (2024)* explored several classifiers such as XGBoost, SVM, and Random Forest and found that combining Named Entity Recognition (NER) with sentence-level N-gram techniques improves the classification performance, particularly in reducing false negatives. In addition, some papers utilized feature detection algorithms such as Oriented FAST and Rotated BRIEF (ORB) algorithms for logo detection and localization (*Bhurtel, Siwakoti & Rawat, 2022*). Studies have focused on balancing data to enhance model accuracy using techniques like SMOTE (*Alsubaei, Almazroi & Ayub, 2024*; *Tamanna et al., 2024*).

Random forest and XGBoost have been used effectively on datasets such as URLs, websites, and job ads, achieving accuracies as high as 98.37% in certain cases (*Devalla et al., 2022*; *Aslam & Nassif, 2023*; *Tamanna et al., 2024*; *Dewan, Kashyap & Kumaraguru, 2014*). Support vector machines (SVM), KNN, and decision trees are also popular classifiers for phishing detection, with varying levels of success, reaching up to 94.87% accuracy (*Shombot et al., 2024*; *Mahbub, Pardede & Kayes, 2022*). Neural networks, particularly BiLSTM and ANN, show significant potential in URL and email phishing detection,

**Table 2  Summary of studies on phishing classification.**

| Method | Data type | Classifier | Accuracy | Citation |
|---|---|---|---|---|
| Deep learning | URL or website | ANN | 90.82% | *Devalla et al. (2022)* |
| | | BiLSTM | 94.41% | *Devalla et al. (2022)* |
| | | DNN | 99.27% | *Bouijij, Berqia & Saliah-Hassan (2022)* |
| | | ResNeXt-embedded GRU | 98.00% | *Alsubaei, Almazroi & Ayub (2024)* |
| | | CNN | 95.76% | *Pires & Borges (2023)* |
| | Emails | LSTM & XGBoost | 98.59% | *He et al. (2024)* |
| Machine learning | URL or websites | Random forest | 95.04% | *Devalla et al. (2022)* |
| | | | 98.37% | *Aslam & Nassif (2023)* |
| | | | 95.00% | *Ito, Takata & Kamizono (2022)* |
| | | XBoost | 94.20% | *Devalla et al. (2022)* |
| | | AdaBoost | 82.36% | *Devalla et al. (2022)* |
| | | KNN | 90.47% | *Devalla et al. (2022)* |
| | | | 94.87% | *Aslam & Nassif (2023)* |
| | | SVM | 84.50% | *Shombot et al. (2024)* |
| | | | 91.49% | *Aslam & Nassif (2023)* |
| | | Random tree | 95.98% | *Aslam & Nassif (2023)* |
| | | Extra-Tree | 98.77% | *Bouijij, Berqia & Saliah-Hassan (2022)* |
| | | Multi-layer perceptron | 96.71% | *Aslam & Nassif (2023)* |
| | Email | Local outlier factor | – | *Wu & Guo (2022)* |
| | | SVM | 98.70% | *Sanchez & Duan (2012)* |
| | | Random forest | 98.28% | *Dewan, Kashyap & Kumaraguru (2014)* |
| | | Decision Tree | 97.32% | *Dewan, Kashyap & Kumaraguru (2014)* |
| | | Naive Bayesian | 69.35% | *Dewan, Kashyap & Kumaraguru (2014)* |
| | | Decision table | 95.05% | *Dewan, Kashyap & Kumaraguru (2014)* |
| | Financial transactions | XBoost & Random Forest | 94.00% | *Tamanna et al. (2024)* |
| | Job ads | Random forest | 91.80% | *Mahbub, Pardede & Kayes (2022)* |
| | | J48 Decision tree | 91.64% | *Mahbub, Pardede & Kayes (2022)* |
| | Financial website | Logistic regression | 97.30% | *Yu et al. (2024)* |
| | | SVM | 97.00% | *Yu et al. (2024)* |
| NLP | Email& Domains | Statistical classifier& NLPRank | – | *Thejaswini & Indupriya (2019)* |
| Clustering | URL or websites | Agglomeration clustering & K-medoids | – | *Zhuang et al. (2012)* |

with accuracies reaching up to 99.27% (*Bouijij, Berqia & Saliah-Hassan, 2022*; *Alsubaei, Almazroi & Ayub, 2024*; *Pires & Borges, 2023*). Heuristic approaches complement these methods by identifying phishing through analyzing sender information (*Sanchez & Duan, 2012*) and applying the local outlier factor on email headers from mirrored SMTP network traffic (*Wu & Guo, 2022*). NLP is explored for detecting various attacks, including phishing emails, through email content and source URL analysis (*Thejaswini & Indupriya, 2019*). Hierarchical clustering and K-medoids have been used to develop automatic categorization systems for grouping phishing websites or malware based on shared characteristics (*Zhuang et al., 2012*). The list of studies with their results are summarized in Table 2.

*Multi-layer classification methods.* Some studies integrate multiple approaches, such as using K-nearest neighbors, and IP-based mechanisms, focusing on IP mapping and logo recognition (*Bhurtel, Siwakoti & Rawat, 2022*), or machine learning algorithms with dynamic firewall rule generation (*Tudosi et al., 2023*). Similarly, the use of the naive Bayesian classifier, fuzzy string comparison, and image hashing results in 95% detection of fake educational domains (*Privalov & Smirnov, 2023*; *Privalov & Smirnov, 2022*). Classifying phishing websites using blacklists alone is not effective as they are not effective against zero-day attacks and cloaked phishing sites (*Oest et al., 2019*; *Chen et al., 2021*). Though, phish Mail Guard integrates blacklist, white list, heuristic techniques, DNS, and textual content analysis for comprehensive phishing email identification (*Hajgude & Ragha, 2012*). *Varshney et al. (2021)* proposed a novel method to uncover DNS over HTTPS traffic for phishing detection. Another study is NoFish, where (*Niroshan Atimorathanna et al., 2020*) combined various mechanisms like URL analysis, visual similarity detection, DNS phishing detection, and an email client plugin. It utilizes machine learning, natural language processing (NLP), and computer vision techniques to detect phishing attacks, achieving an accuracy of 94% for URL detection and 91.67% for email detection. Using information from software-defined networking through deep packet inspection with the help of NNA resulted in an average accuracy of 98.1% (*Chin, Xiong & Hu, 2018*). Heuristics also was used by *Liu & Zhang (2012)*. They proposed two layers of detection where they first compute the weight for the URL features. If the weight does not exceed a threshold, they check the webpage features, providing specialized detection mechanisms for financial phishing.

*Collaborative phishing detection.* Collaboration between organizations can effectively defend against phishing. *Higashino (2019)* designed a system for sharing information about phishing attacks across financial organizations. *Vos, Erkin & Doerr (2021)* presented a privacy-preserving protocol for querying multiple data providers without revealing stored data. Similarly, *Deval et al. (2021)* employed machine learning methods for collaborative phishing detection, allowing for the inclusion of new features in the models while *Salau, Dantu & Upadhyay (2021)* utilized blockchain technology to share data about phishing between organizations. Interestingly, *Stembert et al. (2015)* proposed a method that combines interaction methods to detect email phishing attacks, leveraging the intelligence of both expert users and novice users.

*Authentication based prevention.* Email spoofing techniques such as SPF (Sender Policy Framework), DKIM (DomainKeys Identified Mail), and DMARC (Domain-based Message Authentication, Reporting, and Conformance) help IT staff in organizations to verify the authenticity of the sender's domain to prevent malicious actors and countermeasures phishing emails (*Kulkarni et al., 2024*). However, these protocols have a low adoption rate because of the deployment technical issues, weak incentives, and concerns about blocking legitimate emails (*Hu, Peng & Wang, 2018*), requiring applying other methods to complement them. Similarly, the security multi-factor authentication method–Fast Identity Online 2 protocol was found to be challenging to deploy with concerns of security,

usability, and adaptability in real-world enterprise use cases (*Kepkowski et al., 2023*). Recent studies proposed several authentication solutions, such as the use of dynamic password technology as an alternative to OTP, offer new avenues for preventing phishing attacks (*Xu, Qi & Xi, 2016*) and the brand indicator for message identification to complement the sender's email domain with their organization authenticated logo (*Dolnák & Kampová, 2022*). Additionally, *Bojjagani, Brabin & Rao (2020)* proposed a novel authentication protocol that sends a nonce message to a mobile customer device to avoid phishing attacks. DMARCBox provides analytical reports to combat email phishing, offering accurate graphical reports alongside geolocation mapping of email sources (*Nanaware, Mohite & Patil, 2019*). *Thakur & Yoshiura (2021)* propose AntiPhiMBS-Auth, a model for mobile banking systems, to combat phishing at the authentication level, addressing users who mistakenly download phishing apps and those vulnerable to phishing emails or SMS.

*Other prevention methods.* Other studies focus on specific aspects of phishing attacks. *Teerakanok, Yasuki & Uehara (2020)* presented a practical solution for detecting bogus invoice schemes using checksums from invoices and shared secret information. *Eshmawi & Nair (2019)* aimed to protect organizations from Smishing attacks with a roving proxy framework. *Goel, Sharma & Goswami (2017)* suggested a method to prevent QR code manipulation for sharing sensitive information. 'DeFD' distinguishes disguised executable files in phishing emails transferred over network connections to enhance incident response (*Ghafir et al., 2018*). Since phishing is the first step to initiating other attacks, *Lam & Kettani (2019)* focused on detecting and preventing ransomware delivery through phishing channels, while *Zhang et al. (2012)* introduced a VPN abuse detection system to identify compromised accounts rapidly. Virtual honeypot solutions are also used to prevent access to phishing sites (*Husák & Cegan, 2014*; *Chauhan & Shiwani, 2014*). *Ho et al. (2019)* developed a new detector for lateral phishing attacks to minimize false positives. Insider threat is one of the most common threats for lateral phishing, thus *He et al. (2024)* improved their phishing detection algorithm by utilizing Bi-LSTM with Attention mechanisms to detect insider threats based on user behavior. Since phishers nowadays start to encrypt their websites, *Ohmori (2023)* proposed a method to detect "Let's Encrypt" sites using TLS 1.2 or less as they easily provide free encryption certificates and are commonly used by attackers; however, such tools can be improved to accurately detect malicious websites.

Some solutions aim to protect a specific brand. For example, *Al-Hamar et al. (2021)* proposed a solution for specific organizations to defend against organization-targeted phishing emails, focusing on domain names, while *Ramanathan & Wechsler (2013)* proposed a multi-stage methodology to take down websites impersonating organizations.

These varied approaches underscore the multifaceted nature of phishing defense, highlighting the importance of interdisciplinary collaboration and technological innovation in combating evolving cyber threats.

## Incident response

In response to the evolving landscape and sophistication of phishing threats (*Falowo et al., 2022*), numerous studies have investigated and developed mitigation strategies to combat phishing attacks and mitigate their damaging impacts.

*Incident analysis and management.* Incident analysis is an important step in the fight against phishing because organizations are required to mitigate the attack and learn from it to prevent future attacks. *Lacey, Salmon & Glancy (2015)* conducted a system analysis of processes, documentation, and activity logs to explore cyber situational awareness in organizations. To gain a deeper understanding of attack impact and the organizations' response, tools for analyzing and visualizing phishing attacks hitting the organizations have been developed to enhance cyber situational awareness of targeted phishing attacks (*Legg & Blackman, 2019*) and to assess security risks in banking institutions (*Gupta et al., 2021a*). Furthermore, *Gupta et al. (2021b)* developed a conceptual model to investigate social engineering attacks in a calculated way from several perspectives including attacker methods, exploit weaknesses, and the consequences of the attack on the cryptographic algorithms. Similarly, *Lohiya & Thakkar (2024)* reviewed several attack modeling techniques (*e.g.*, cyber kill chains, the diamond model, and security incident response matrix) for phishing attacks, revealing the significance of models to improve the risk assessment and response strategies. These tools and models help network defenders better understand the attacker's methodology, assess the risks at each attack stage, and implement timely mitigation strategies. Using a survey of professionals from major accounting firms in Nigeria, the research finds that digital forensic accounting significantly helps in reducing phishing scams, advance fee fraud, and credit card fraud (*Awodiran et al., 2023*).

*Phishing reporting.* Reporting phishing is essential in the fight against phishing because it allows organizations to respond quickly and block the attacks before they cause harm. To enhance the reporting rate, *Burda, Allodi & Zannone (2020)* proposed a crowd-sourced approach to automate response and containment against spear phishing, empowering users and strengthening resilience, which is observed by *Lain, Kostiainen & Čapkun (2022)* who found that employees can effectively and quickly report phishing emails while maintaining consistent reporting rate. Interestingly, while 31% of employees had the intention to report phishing emails, the most believable phishing emails are less likely to be reported compared to obvious ones (*Kersten et al., 2022*), making it important to raise awareness about the importance of reporting. Interestingly, this approach has shown to be successful in a small and medium-sized organization because of the strength of the community as employees are encouraged to share suspicious communications quickly, which can significantly enhance phishing resilience (*Burda et al., 2023*). For example, it can reduce the cost of advanced security plans by utilizing the human firewall system in detecting and preventing phishing attacks (*Shin et al., 2023*). *Althobaiti, Jenkins & Vaniea (2021)* conducted a case study on the phishing response procedures to understand how phishing reports are handled in a UK-based university and found that the number of reports can be unmanageable even though the percentage of reports is low compared to the size of the organization; therefore, *Althobaiti et al. (2023)* proposed a clustering approach that aims to group similar emails into campaigns for the IT teams to deal with them.

*Learning from previous attacks.* Current attacks were examined by several studies to help organizations learn from incidents to prevent future events.

*Oest et al. (2018)* explored the anti-phishing ecosystem through phishing kit analysis, seeking insights into countering evolving social engineering techniques employed by cybercriminals. Additionally, profiling phishing emails based on attack groups has assisted organizations in understanding attack motives and devising effective countermeasures (*Lee et al., 2021a*). Using clustering techniques, *Vargas et al. (2016)* investigated the registered attack on a financial US institution by grouping phishing websites based on their similarities to distinguish attacker groups. Their findings can be utilized to update the anti-phishing filters to prevent such tactics. Other studies examined the human factors that are exploited by attackers, such as the use of targeted phishing that increases the success rates in penetrating organizational defenses, as seen in Swedish organizations (*Holm et al., 2014*), a telecommunication organization (*Abdullah & Mohd, 2019*), a university, and a large international consultancy company (*Burda et al., 2020*). This was also observed by *Kotson & Schulz (2015)* who found that phishers send unique curriculum vitae (CV) attachments based on the target victims' profiles. Furthermore, the way the phishing attack was delivered plays a significant role, sometimes more than the content of the message itself (*Burda et al., 2020*). The source of an email is one of the tactics used to deceive victims; for instance, emails spoofing an information technology (IT) department have led to a higher percentage of compromised accounts in an Emirates university (*Mohebzada et al., 2012*) and Swedish organizations (*Holm et al., 2014*). Similarly, showing professionalism in emails or phone calls that spoof banks is one of the tactics as they resemble the messages that they usually receive from their banks (*Jansen & Leukfeldt, 2015*). This tendency was also observed with Jordanian students who click on links from seemingly familiar sources, such as friends or relatives (*Manasrah, Akour & Alsukhni, 2015*). While these tactics can combat phishing attacks, frequent analysis of attack messages and delivery methods is needed, as a recent study found a significant increase in attack sophistication from 2010 to 2023. For example, email topics are shifted from security-focused to campus life topics along with a reduction in spelling errors (*Morrow, 2024*).

Cognitive vulnerabilities such as authority, liking, scarcity, consistency, social proof, and reciprocity were also exploited by phishers (*Taib et al., 2019*; *De Bona & Paci, 2020*; *Williams, Hinds & Joinson, 2018*). However, these strategies do not always guarantee success as a phishing simulation in a multinational financial organization showed that while some users fell for authority-based lures, scarcity was perceived as the least credible tactic (*Taib et al., 2019*). Social proof is also a powerful tool, as people are more likely to trust a source that appears to be trusted by others (*Taib et al., 2019*). Social distance can also be exploited as the more individuals perceive similarity with the sender, the higher the trust and the greater the risk of data compromise (*Martin, Lee & Parmar, 2021*). *Van Der Heijden & Allodi (2019)* analyzed cognitive vulnerabilities in phishing attacks to prioritize remediation efforts based on vulnerability triggers, to predict users' behavior and effectively mitigate the impact of phishing campaigns whereas *Abroshan et al. (2021)* presented a phishing mitigation solution leveraging human behavior and emotional cues to identify high-risk

users and apply appropriate mitigation strategies. This system evolves to provide tailored protection, enabling organizations to effectively safeguard vulnerable users.

*Attack notifications.* After a successful build of a phishing detection system, *Pires & Borges (2023)* developed a phishing responder model that does at least one of the following to the detected phishing attack: reporting the website *via* email, notification post on a Telegram channel and automatic reporting to Google SafeBrowsing. Sending warning emails about ongoing attacks can have limited effect as a preventive measure. For example, despite receiving warning emails from the IT department, some individuals still fell victim to phishing attacks, as noted by *Mohebzada et al. (2012)* and *Holm et al. (2014)*. Similarly, management messaging does not appear to directly influence outcomes, as warning messages have not been observed to reduce the number of clicks in phishing simulations (*McElwee, Murphy & Shelton, 2018*). These findings suggest that alternative or supplementary strategies may be necessary to prevent phishing attacks and mitigate their impact on organizations effectively.

These diverse approaches underscore the multifaceted nature of responding to phishing attacks, highlighting the importance of integrating technological innovations with insights from human behavior and cognitive psychology to develop comprehensive anti-phishing strategies.

## GAPS AND OPEN QUESTIONS

### Frameworks have advantages and disadvantages

Our investigation revealed that organizations frequently rely on established frameworks for IT and security management. These frameworks are guidelines designed to help organizations identify problems and adapt the practices and procedures based on their needs. While these guidelines offer flexibility, their lack of specificity makes it challenging for organizations to adopt them successfully (*Stevens et al., 2022*). The organization's sector, size, and budget play a significant role in tailoring practices, sometimes making it almost impossible to implement the frameworks or policies. This gap highlights the need for additional research that focuses on identifying these challenges based on the mentioned factors and provides a list of recommendations with the pros and cons of each, allowing stakeholders to decide which recommendation to follow. Identifying these challenges can also help researchers to develop tool-based solutions that simplify adherence to the guidelines.

### Phishing can still pass through

Our literature assessment reveals continuing gaps and unsolved concerns that potentially provide opportunities for phishers to exploit systems, despite significant efforts in research and practical interventions to prevent phishing emails. Due to the nature of phishing attacks, organizational phishing management requires a multi-layer defense system starting from preventing the attack to mitigating the impact of the harvested victims and learning from that incident. Although this research demonstrates the substantial progress made in preventative measures that incorporate human and technical variables,

it also identifies a notable lack of studies on post-attack strategies. Research indicates that companies frequently lack the resources necessary to quickly and efficiently react to phishing attacks, which highlights the critical need for further investigation into post-attack recovery and defense mechanisms (*Naqvi et al., 2023*) to ensure comprehensive protection against phishing threats; for example, developing tools that can remove phishing emails from users inboxes as a replacement for the ineffective attack notification messages or developing tools that can update the blocking filters immediately for ongoing attack. Furthermore, longitudinal studies tracking phishing incidents and organizational responses over time could offer valuable insights into evolving trends and effective countermeasures in organizations. Learning that attackers change their tactics frequently, invitation studies, whether longitudinal or not, can benefit from automating the studies to make it easy for organizations and researchers to repeat the investigation when needed. Unresolved phishing attacks can lead to other and more damaging security issues, such as lateral phishing, where emails are sent from legitimate accounts (*Ho et al., 2019*). Future research should not only be on reducing the negative impact of phishing that passes through but also on detecting and preventing other attacks and training employees to recognize them before falling victims (*Chitare, Coventry & Nicholson, 2023*).

## Shift of common attack vectors

The literature examined in this study encompasses a diversity of phishing vectors, including Email (60 occurrences), websites (19), URLs (10), QR codes (1), social networks (1), mobile web apps (3), SMS (1), and telephone calls (1). Email is the most researched phishing vector, with URLs and websites coming in second and third. The frequency of these vectors emphasizes the importance of investigating how to protect against them and comprehending how vulnerable people are to their social engineering tactics, which seek to encourage activities such as opening links, downloading malicious files, or disclosing private information. Additionally, researchers have made attempts to increase user knowledge of these attack vectors.

Nonetheless, there has been a noticeable increase in mobile phone-based phishing attacks, known as vishing. Although attackers have historically preferred emails with embedded links (*Verizon, 2022*; *APWG, 2023*; *ProofPoint, 2023*) reported an increase in vishing attacks- 40% in 2023 as compared to 2022- emphasizing the need for more research on this evolved threat vector. For example, *Jansen & Leukfeldt (2015)* discovered that victims frequently consider telephone-based attacks to be authentic since attackers simply need only to seem trustworthy. Similarly, there is a rapid increase in QR-code-based phishing attacks that deliver malicious links or attachments (*ProofPoint, 2024*). Organizational-based studies that target QR-Codes are limited although this attack vector is particularly dangerous. It is impossible to recognize phishing just by looking at the QR Code itself.

More research is necessary to thoroughly examine the aforementioned attacks and create practical detection and mitigation techniques, given the dynamic nature of phishing attempts.

## Impact of sectors on phishing management

In our literature review, we observed a significant number of studies that focus on specific sectors, with educational organizations being the most investigated (25 occurrences), followed by financial institutions (15 occurrences), and industry/manufacturing sectors (10 occurrences). The comparative analyses across these sectors shown by several studies revealed that different sectors exhibit varying levels of susceptibility to phishing attacks and implement diverse security measures.

Further research can explore how organizations develop and implement phishing management plans specific to their sector. For instance, studies could investigate the effectiveness of sector-specific phishing awareness training programs and the adoption of security measures tailored to the particular risks faced by each sector. Additionally, comparative studies across sectors could highlight the unique challenges and best practices in phishing prevention and mitigation strategies. Given that financial organizations are frequent targets of phishing attacks, in-depth investigations into the tactics and strategies employed by attackers targeting these sectors could provide valuable insights for improving organizational defenses.

## Methodological barriers in phishing studies

Phishing simulation is one of the most utilized methodologies in phishing for testing users' awareness, training users, and understanding their susceptibility to phishing. However, there are ethical considerations surrounding the use of such exercises. Such studies require informed consent about the purpose and risks of the simulation and a safe strategy to deliver the attack to users (*Finn & Jakobsson, 2007b*; *Finn & Jakobsson, 2007a*).

In addition to ethical concerns, the procurement process of implementing phishing simulations often reveals hidden costs that are typically overlooked (*Brunken et al., 2023*). While much of the existing research focuses on measuring user behavior through click rates or other immediate reactions, it often neglects the significant time and effort required from various organizational stakeholders, including IT, legal, and HR departments. These hidden costs can be a barrier, particularly for smaller organizations, making the deployment of simulations more challenging than anticipated. Challenges such as stakeholder involvement, technical difficulties, and system integration create friction in the process, often hindering successful implementation.

Most of the research done on organizations is typically carried out with the assistance and collaboration of actual IT departments. This collaboration ensures that the experiments are conducted in a controlled and ethical manner while ensuring that appropriate safeguards are in place to protect employees and organizational assets and facilitate the experiment procedures. Further study of the collaboration between organizational stakeholders and researchers is essential for advancing research in the area and addressing the methodological barriers posed by hidden costs, procurement challenges, and simplistic evaluation methods.

## Generative AI in organization-based phishing research

Generative AI (GenAI) models, such as ChatGPT, present both risks and opportunities in organizational phishing research. While GenAI tools are primarily designed to assist with

generating human-like text, attackers can exploit them to create sophisticated phishing emails that mimic legitimate communication styles, making detection difficult (*Gupta et al., 2023*). Attackers can use techniques such as reverse psychology to manipulate these models, bypassing ethical constraints and generating cyber threats, including phishing attacks and malware. This misuse of AI highlights the need for research directed at understanding the capabilities of chatbots and safeguarding solutions to prevent exploitation by adversaries.

However, GenAI tools can also be leveraged defensively to enhance security within organizations. These AI models can analyze large amounts of data to detect patterns and anomalies that indicate phishing attempts, providing early threat detection (*Shanthi, Sasi & Gouthaman, 2023*). Additionally, they can automate incident response processes, helping organizations respond more quickly to phishing attacks by reducing the manual workload on security teams. Tools powered by AI can also assist in vulnerability management by identifying and prioritizing weak points in the organization's systems that could be exploited by phishers. Despite these benefits, researchers should explore the possibility of utilizing chatbots in research while focusing on the challenges of combining this use such as data quality issues, model explainability, and potential bias (*Shanthi, Sasi & Gouthaman, 2023*).

## LIMITATIONS

A potential limitation of our study is that we used several organization-equivalent keywords to identify relevant research that targets organizations. While this approach could have resulted in the omission of some papers, we minimized this risk by using keywords commonly found in recent studies and thoroughly searching for them in the title, abstract, and keywords sections. Additionally, the potential for bias in selecting and analyzing the literature is acknowledged, as is the possibility that our classification of some studies may differ from the author's original intent. To mitigate these concerns, we conducted multiple discussions and iterations throughout the analysis process, ensuring a more balanced and comprehensive review.

## CONCLUSION

Phishing remains an evolving threat to organizations around the globe. Its effective defense requires extensive cybersecurity measures. Although prior work provided insightful interventions, analytical data, and patterns, a multi-layered strategy that addresses prevention, detection, and mitigation is required. Developing successful phishing prevention techniques will require ongoing cooperation and innovation. By filling up the gaps in the literature and expanding our knowledge on phishing, we can enhance organizational security and mitigate the impact of cyber threats.

## APPENDIX. DIGITAL LIBRARIES QUERIES

Queries were executed on the 13th of April, 2022.

### IEEE Xplore search query

((("All Metadata":Phishing AND "All Metadata":organisation) OR ("All Metadata": Phishing AND "All Metadata":organization) OR ("All Metadata": Phishing AND "All Metadata":Institution) OR ("All Metadata": Phishing AND "All Metadata":corporation) OR ("All Metadata": Phishing AND "All Metadata":enterprise) OR ("All Metadata": Phishing AND "All Metadata":workplace) OR ("All Metadata": Phishing AND "All Metadata":organisational) OR ("All Metadata": Phishing AND "All Metadata":organizational) OR ("All Metadata": Phishing AND "All Metadata":incident) ("All Metadata":Phish AND "All Metadata":organisation) OR ("All Metadata": Phish AND "All Metadata":organization) OR ("All Metadata": Phish AND "All Metadata":Institution) OR ("All Metadata": Phish AND "All Metadata":corporation) OR ("All Metadata": Phish AND "All Metadata":enterprise) OR ("All Metadata": Phish AND "All Metadata":workplace) OR ("All Metadata": Phish AND "All Metadata":organisational) OR ("All Metadata": Phish AND "All Metadata":organizational) OR ("All Metadata": Phish AND "All Metadata":incident))

### ACM search query

[[Title: phishing] OR [Title: phish] OR [Abstract: phishing] OR [Abstract: phish] OR [Keywords: phishing] OR [Keywords: phish]] AND [[Title: organisation] OR [Title: organization] OR [Title: institution] OR [Title: cooperation] OR [Title: enterprise] OR [Title: incident] OR [Title: workplace] OR [Title: organisational] OR [Title: organizational] OR [Abstract: organisation] OR [Abstract: organization] OR [Abstract: institution] OR [Abstract: cooperation] OR [Abstract: enterprise] OR [Abstract: incident] OR [Abstract: workplace] OR [Abstract: organisational] OR [Abstract: organizational] OR [Keywords: organisation] OR [Keywords: organization] OR [Keywords: institution] OR [Keywords: cooperation] OR [Keywords: enterprise] OR [Keywords: incident] OR [Keywords: workplace] OR [Keywords: organisational] OR [Keywords: organizational]]

### ScienceDirect.com search query

title, abstract, keywords: ((phishing OR Phish) AND (organisation OR orgranization OR Institution OR cooperation OR Enterprise OR incident OR workplace OR organisational OR organizational OR Incident))

### Funding

This work was funded by the Deanship of Scientific Research, Taif University. The funders had no role in study design, data collection and analysis, decision to publish, or preparation of the manuscript.

### Grant Disclosures

The following grant information was disclosed by the authors:
The Deanship of Scientific Research, Taif University.

## Competing Interests

The authors declare there are no competing interests.

## Author Contributions

- Kholoud Althobaiti conceived and designed the experiments, performed the experiments, analyzed the data, authored or reviewed drafts of the article, and approved the final draft.
- Nawal Alsufyani conceived and designed the experiments, performed the experiments, analyzed the data, authored or reviewed drafts of the article, and approved the final draft.

## Data Availability

This is a literature review.

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
