# Peer review of "A review of organization-oriented phishing research"

_PeerJ Computer Science, doi:10.7717/peerj-cs.2487_

## Round 0.1 · original submission · Major Revisions

To improve the manuscript's contribution to the field, significant revisions are needed to address highlighted weaknesses. Incorporating figures and charts will summarize existing works using visual elements to enhance comprehension and presentation. Updating the literature review with the latest studies from 2023 will ensure its comprehensiveness. Clarifying selection criteria by providing a detailed discussion of the criteria used for selecting papers will enhance the robustness and reproducibility of the methodology. Restructuring the results section will organize the findings in a more coherent and progressively logical manner. Conducting sampling for missing rates will evaluate the current keyword-based method's completeness by providing a reasonable missing rate. Tabulating findings in tables categorized by taxonomies will benefit readers and reviewers. Providing deep insights through a detailed discussion of the advantages and disadvantages of various frameworks and suggesting methods to fill current research gaps will enhance the manuscript. Enhancing comparative analysis by adding more data and conducting a thorough comparison of phishing detection techniques will increase the review's utility for researchers. Finally, correcting typographical errors will improve the manuscript's readability.

Reviewer 1 ·

Basic reporting

The paper is generally well-written, using clear and professional language.
The paper discussed the importance of organization-oriented phishing research, and reviewed the breadth of existing works, which provides the sufficient background.
The review is within the scope of the journal.
It makes it clear who the audience is and the motivation of this work.
But one weakness of the work is the lack of figures and chart. It will be better to use figures or charts to summarize exisitng works.

Experimental design

The paper analyzed works between 2012 and 2022 based on predetermined criteria.But it does not include the latest work, such as 2023. Also the criteria is not well disucussed, and this should be more explict.
The sources are adequately cited.
Logical Organization: The paper is organized logically; however, some subsections under the results could be restructured to ensure that findings are presented in a more coherent and progressively logical manner.
The paper is organized logically.

Validity of the findings

The paper is well developed. The paper successfully identifies gaps in current reserch and suggests future direction. s.

Cite this review as
Anonymous Reviewer (2024) Peer Review #1 of "A review of organization-oriented phishing research (v0.1)". PeerJ Computer Science

·

Basic reporting

no comment

Experimental design

Authors used keywords to search for all relevant paper. This method does not sound scientific to me. This might involve a lot of bias and incompleteness. I appreciate that authors have put their queries in the appendix, I want to see how many relevant paper will be missing by using current keywords. I suggest author to do a quick sampling, providing us with a reasonable missing rate. Otherwise, this method is too ad-hoc.

Validity of the findings

The findings of this paper were presented all in text, which are easy to catch, and reference. Can you tabulate your findings by taxonomies? This will benefits future readers, and help reviewers to justify your contributions.

Cite this review as

Reviewer 3 ·

Basic reporting

This paper tries to summarize existing literatures related to organization phishing research from 2012 to 2022 (not including the most recent ones from 2023 is indeed a concern)

Experimental design

The authors put a lot of papers of reference into this paper with some organization. However, the flow seems to be only tediously listing all the papers found and summarize each one by one with some categorization.

Validity of the findings

Authors summarized a few potential gaps and open questions. But some of them seem to not provide many insights from the authors. For example, one findings of the authors is "Frameworks have Advantages and Disadvantages" which seems to be very general. However, I was wishing to see the authors could discuss more about what are the most important advantages and disadvantages. But this discussion is not included thus hindering the contribution of the paper.

Additional comments

I want to thank the authors for submitting their work. However, I have several concerns for this paper.

1. my major concern is the lack of insights of this survey. I was hoping that the authors could provide more insights about what is the most important and interesting research directions about the organization-oriented phishing. I wanted to see more insights from the authors about what methods can be potentially used to fill the current gaps. Unfortunately, the authors only list a few very general statements about the gaps without their own insights in this area.

2. Another major concern is the readability of this paper. The paper has no table, figure, etc, with full words. It's very hard to follow it since it only lists few sentences of summaries of each paper one by one.

Cite this review as
Anonymous Reviewer (2024) Peer Review #3 of "A review of organization-oriented phishing research (v0.1)". PeerJ Computer Science

Reviewer 4 ·

Basic reporting

The manuscript provides a comprehensive summary of existing literature on phishing, reviewing 106 papers and focusing on organization-oriented research. It effectively categorizes current studies and identifies future opportunities for exploration. However, while the overview of different phishing detection techniques is adequate, the paper lacks a detailed comparative analysis of these methods. For example, in protection, you can list some data and make a better clarification. A more thorough comparison could enhance the utility of this review for researchers.

Additionally, there are typos, lines 67-68, 169-170, and 536-538.

Experimental design

I have no concerns about the study design.

Validity of the findings

I have no concerns about the validity of the findings.

Cite this review as
Anonymous Reviewer (2024) Peer Review #4 of "A review of organization-oriented phishing research (v0.1)". PeerJ Computer Science

---

## Round 0.2 · accepted · Accept

Thanks! The reviewers are happy with your revision and congratulations!

Reviewer 1 ·

Basic reporting

no comment

Experimental design

no comment

Validity of the findings

no comment

Cite this review as
Anonymous Reviewer (2024) Peer Review #1 of "A review of organization-oriented phishing research (v0.2)". PeerJ Computer Science

·

Basic reporting

This paper concludes with the most up-to-date research on phishing, offering comprehensive insights into phishing campaigns and detection methods. It will be highly beneficial for future research in the long run.

Experimental design

I appreciate the authors explaining their keyword search method in the rebuttal. I’m satisfied with the current approach, provided it includes up-to-date research. I noticed that the authors have incorporated research up to 2024, which significantly enhances the paper.

Validity of the findings

I believe the author has addressed my concerns. The tabulation of references improves readability and paper structure.

Cite this review as

Reviewer 3 ·

Basic reporting

N/A

Experimental design

N/A

Validity of the findings

N/A

Additional comments

The authors fixed the issues of lack of latest literature, restructured the flow of summaries, and also included additional discussion on how to defend against the phishing techniques.

Overall I think the authors addressed most of my concerns and also comments from other reviewers.

Cite this review as
Anonymous Reviewer (2024) Peer Review #3 of "A review of organization-oriented phishing research (v0.2)". PeerJ Computer Science

Reviewer 4 ·

Basic reporting

Thanks for the revision. The manuscript and response letter addressed my concerns. I would recommend accepting the manuscript.

Experimental design

I have no concerns about the study design.

Validity of the findings

I have no concerns about the validity of the findings.

Cite this review as
Anonymous Reviewer (2024) Peer Review #4 of "A review of organization-oriented phishing research (v0.2)". PeerJ Computer Science